# Optimal User Scheduling in Multi Antenna System Using Multi Agent Reinforcement Learning

**DOI:** 10.3390/s22218278

**Published:** 2022-10-28

**Authors:** Muddasar Naeem, Antonio Coronato, Zaib Ullah, Sajid Bashir, Giovanni Paragliola

**Affiliations:** 1Institute of High Performance Computing and Networking, National Research Council of Italy, 80131 Naples, Italy; 2Centro di Ricerche sulle Tecnologie ICT per la Salute ed il Benessere, Università Giustino Fortunato, 82100 Benevento, Italy; 3Department of Electrical Engineering, National University of Sciences & Technology, Islamabad 44000, Pakistan

**Keywords:** reinforcement learning, user scheduling, channel capacity, MIMO, MU-MIMO, next-generation networks, fairness, sumrate

## Abstract

Multiple Input Multiple Output (MIMO) systems have been gaining significant attention from the research community due to their potential to improve data rates. However, a suitable scheduling mechanism is required to efficiently distribute available spectrum resources and enhance system capacity. This paper investigates the user selection problem in Multi-User MIMO (MU-MIMO) environment using the multi-agent Reinforcement learning (RL) methodology. Adopting multiple antennas’ spatial degrees of freedom, devices can serve to transmit simultaneously in every time slot. We aim to develop an optimal scheduling policy by optimally selecting a group of users to be scheduled for transmission, given the channel condition and resource blocks at the beginning of each time slot. We first formulate the MU-MIMO scheduling problem as a single-state Markov Decision Process (MDP). We achieve the optimal policy by solving the formulated MDP problem using RL. We use aggregated sum-rate of the group of users selected for transmission, and a 20% higher sum-rate performance over the conventional methods is reported.

## 1. Introduction

Higher data rates and reliability are a few challenges faced by 5G and beyond technologies. MIMO systems are becoming one the significant pillars in wireless communication due to their spectral efficiencies and diversity gains [1]. In addition, MU-MIMO systems further exploit both spatial multiplexing and spatial diversity for reliable communication links and higher data rate [2]. MU-MIMO systems performance is superior to that of single-user MIMO communication networks due to its ability to serve more than one user in a Transmission Time Slot (TTS) and frequency band [3].

Moreover, MU-MIMO systems provide a notable advantage over conventional communication systems (e.g., its multiplexing gain is proportional to the number of transmit antennas, even though the user device doesn’t need to have many antennas). This minimizes the burden because of limitations of size and user equipment cost [3]. The other benefit is a fewer impact on propagation problems, i.e., antenna correlation and channel rank, since multi-user diversity address propagation issues. MU-MIMO is on the other hand, sensitive to Channel Stae Information (CSI) accuracy due to inter-user interference and may be mitigated as in [4,5].

Recent advances in Depp Neural Network (DNN) have made RL [6] the most important and attractive Artificial Intelligence (AI) technology. RL is a machine learning branch where an agent interacts with a given environment (mostly unknown), chooses actions, and gradually explores the environment’s characteristics. RL and DNN have been used in diverse research and real-life areas such as computer vision, the configuration of resources, self-organized systems, games, natural language processing, communication and networking, robotics, and autonomous control [7,8,9,10,11].

Machine Learning (ML) techniques have been used recently in different aspects of communication systems, i.e.,  Resource Allocation (RA). For example, deep learning has been applied in [12] for RA in massive MIMO systems. Similarly, applications of RL in RA management of MIMO systems have been found [13,14,15,16], but RL-based methods have not been used for scheduling multiple users in uplink MU-MIMO systems.

The proposed work aims at investigating the potential use of RL in MIMO communications with attention on MU-MIMO systems. The motivation behind this work stems from relevant aspects, such as (1) the importance of allocating radio resources using a scheduling mechanism at the base station in future wireless communication networks; (2) the limitations of existing state-of-the-art user scheduling techniques; and (3) the robustness of RL methods in alleviating these shortcomings and giving an efficient performance.

Moreover, the RL, being one of the most appealing technology, learns the dynamics of an environment MU-MIMO scheduling in this case) with its experience. One does not need to set values of different parameters as required in standard scheduling algorithms. Instead, the agent can learn the optimal combination of wireless channel parameters and optimally select the group of users out of all candidate users for transmission due to its decision-making ability.

The paper is organized as follows. Section 2 provides the state-of-the-art on user scheduling in MU-MIMO systems. Section 3 reports a brief overview of the RL methodology and an introduction to MIMO communication. In contrast, Section 4 provides a detailed introduction to the proposed approach with a problem statement, problem formulation, and RL scheduling algorithm. The results are presented in Section 5. Finally, the concluding remarks are given in Section 6.

## 2. Related Work

Optimal use of radio resources is essential to enhance the system capacity, and user schedule can play a crucial role [17]. We reviewed several user selection schemes adopted in the literature using various selection mechanisms. These methods can be broadly categorized into three types. The first group of tools adopts a certain system parameter for the selection of users (e.g., include SLNR and SINR, etc.) [18]. The second category of the scheduling algorithm considers the scenario where the CSI is not available at the Base Station (BS). The third technique addresses the issue of fairness; i.e., consider fairness as the only performance metric to ensure the quality of service [19].

From the game theory perspective, a user-centric access point scheduling for cell-free massive MIMO systems has been investigated in [20]. Authors have developed a user-centric access-point cluster model as a local altruistic game. Moreover, a maximum non-neighbor-set-based concurrent spatial adaptive play technique is to obtain the Nash equilibrium.

Similarly, a user-selection mechanism for MU-MIMO systems in uplink mode is presented in [21], where the authors used antennas and the ZF detector at the receiver in the BS. They consider the scenario of imperfect channel estimation with AWGN and Rician fading channels. The objective of the user selection is to maximize SNR.

Another user scheduling framework for a cooperative nonorthogonal multiple access scenario is developed in [22]. Deep learning technology has been employed to recognize and classify the channels of imperfect CSI. Deep learning was used to enhance the accuracy of CSI. While authors in [23] consider an end-to-end design of MU-MIMO systems in a downlink scenario, including precoding, limited feedback, and pilot sequences. Then, DL method is used to jointly optimize the precoder design at a base station BS and generate user feedback information. The neural network is used at BS to produce pilot sequences and assists the users in obtaining CSI.

A beam-user selection based on machine learning and low complexity hybrid beamforming infrastructure for the multiuser massive MIMO downlink system is presented in [24]. The householder reflectors are employed to produce the orthogonal analog beamforming matrix. The proposed scheme also uses a feedforward neural network and shows reasonable performance in terms of energy efficiency in the ill-conditioned massive MIMO environment, while the joint user selection and optimal transmit power and antenna selection have been discussed in [25]. The problem of joint user selection and optimal transmit power, and antenna selection is formulated to address inter-cell interference in multi-cell massive MIMO networks. A novel power consumption technique is also used to analyze precise power consumption.

The problems of max-product and max–min power allocation have been formulated in [26,27] by using SINR and SLNR mechanism for linear precoder design. DNN is deployed to predict the optimal power allocation based on each user’s location and helps to minimize the system’s processing time in identifying the optimal power allocation.

An SLNR-based user scheduling approach is presented in [28], where a user’s leakage power to other users is considered the major parameter to decide whether the user should be chosen. Another similar approach is proposed in [29] that also addresses user selection. A modification to the leakage-based method regarding the selection of the first user was presented in [30]. Block diagonalization-based technique is proposed in [31], and the authors claim to achieve reasonable fairness and capacity among users. The authors of [2] developed a data detection receiver and joint maximum likelihood modulation classification of the co-scheduled users. In [32], the authors have considered fairness and sum-rate performance metrics by proposing a near-optimal scheduling algorithm.

A resource allocation mechanism is developed in [33] by using a POMDP method for downlink transmit beamforming at BS equipped with multi-antennas. The authors have used the myopic policy in designing the scheme to prevent the high computational complexity of the value iteration technique. A binary FPA is used in [34] for both antenna and user scheduling to obtain sum-rate performance with reduced computational complexity. An ML-based joint infrastructure for a hybrid precoder and user scheduling is proposed in [35] for MU-MIMO to improve the sum rate. The first component is based on cross-entropy, while the latter is based on the Correlation factor. Keysight’s electronic system-level software is used to produce a channel matrix.

We have reviewed the diverse types of related works. Some works consider the standard parameters for user selection to enhance system capacity, while others apply deep learning and machine learning methods for resource or power allocation. The proposed work is the first concerning using Reinforcement Learning to solve the optimal scheduling in MU-MIMO systems.

## 3. Technical Background

In this section, we report an introduction to RL and to MIMO communication.

### 3.1. Reinforcement Learning

The goal of an RL agent is to interact with a given environment and learn the dynamics of that environment as demonstrated in Figure 1. In the learning process, an RL agent can take action $a_{t}$ out of available actions set in any state $s_{t}$ at *t* time interval and then receive a corresponding reward. After much trial and error, the agent learns the best action for each environment state. The process of learning optimal action for every state forms an optimal policy. The first step is to formulate an underlying problem into an MDP problem and then use a suitable RL technique to solve the problem.

After selecting an appropriate RL method for a modeled MDP, the next step is keeping an exploration-exploitation balance. The RL agent has the option either to use known rewarding actions or can find new actions that may prove more beneficial. So, only one scheme may not be a good strategy, and an RL agent must be able to learn a trade-off between the two.

RL algorithms may be categorized as value-based, policy-based, or a combination of both (e.g., the actor-critic algorithm). This classification can be done between model-free and model-based techniques. The first group of algorithms concerns those techniques that do not require a precise model of the environments to be controlled, such as the Q-learning method as given in Algorithm 1. At the same time, the second class of methods exploits the model and provides an analytical solution to the MDP that describes the environment, such as dynamic programming. Dynamic programming has two variants, i.e., value iteration and policy iteration. The former is summarized in Algorithm 2.

Therefore, the first class of tools, thus, rely on trial and error to update their knowledge and experience about the given environment. The RL agent has to interact with the environment repeatedly to learn the environment. A few examples of this category are the temporal difference and Monte Carlo.
**Algorithm 1:**Q-learning algorithm.1. InitializeQ arbitrarilyQ (terminal) =0Repeat      initialize s      Repeat            choose $a^{\prime }\in \epsilon -greedily$            take action a, observe $r,s^{\prime }$            $Q\left(s_{t},a_{t}\right)\leftarrow Q\left(s_{t},a_{t}\right)+\alpha [r_{t+1}+\gamma Q\left(s_{t+1},a_{t+1}\right]-Q\left(s_{t},a_{t}\right)$            $s\leftarrow s^{\prime }$      s is terminaluntil convergence

**Algorithm 2:**Value iteration algorithm.
Initialize *V* arbitrarily
Repeat
      Δ←0
      For each s∈S
            v←V(s)             $V\left(s\right)\leftarrow max_{a}\sum_{s^{\prime },r}p\left(s^{\prime },r|s,a\right)\left[r+\gamma V\left(s^{\prime }\right)\right]$
            Δ←max(Δ,|v−V(s)|)
until Δ<θ (a small positive number)
output a deterministic policy, π, such that


$\pi \left(s\right)=argmax_{a}\sum_{s^{\prime },r}p\left(s^{\prime },r|s,a\right)\left[r+\gamma V\left(s^{\prime }\right)\right]$




In our problem, first, each user selects a resource block and sends this information to BS so that BS can choose a group of users based on the received information. We modeled a resource block as a single state MDP, and Bayesian tools are suitable to solve such problems. The probability distribution for RV *X* is employed to make an inference about an RV *X* in Bayesian RL, and later extraction on distribution is done for inferences [36]. Such a process is performed by following these steps:Take a prior distribution P(X);P(X) is the belief about RV *X* with no data observation;take a statistical model P(Y|X);P(Y|X) is the statistical dependence and belief about RV *Y* given the *X*;Make observation on data Y=y;Find the posterior distribution using the Bayes rule as in [36].
(1)$$P\left(X|Y=y\right)=\frac{P(y|X)P(X)}{\int P\left(y|X^{{}^{\prime }}\right)P\left(X^{{}^{\prime }}\right)dX^{{}^{\prime }}}$$

### 3.2. MIMO Communication

MIMO is different from single antenna systems as data transmission, and reception in MIMO is done on multiple antennas. Moreover, MIMO introduces signaling degrees of freedom, also known as the spatial degree of freedom, and it is absent in single antenna systems [37]. Exploiting spatial degrees of freedom may be done for “multiplexing”, “diversity”, or a combination of both.

A communication system with a transmitter with many antennas and a receiver with many antennas is considered a single-user MIMO system. Similarly, a communication system with a single transmitter, but many devices on the other side, each with one or more antennas, is categorized as MU-MIMO system. More introduction to MU-MIMO is given in Section 4.

Another important extension of MIMO is the massive MIMO framework. It may be considered a massive MIMO network consisting of more than one MU-MIMO infrastructure. This type of network employs three concepts of beamforming, spatial multiplexing, and spatial diversity.

Precoding is a key process in MIMO employed to map *K* data streams to $Tr_{x}$ transmitting antennas. In single-user MIMO, *K* data streams belong to single-user, but in MU-MIMO *K* data streams are intended for *K* users with a single symbol per user. Some well-known precoding techniques are SVD precoding or optimal unitary precoding, codebook-based precoding, ZF precoding, and DPC. The first three are linear, while DPC is a non-linear coding method.

## 4. System Model

Before explaining the system model’s main components, such as the problem statement, its formulation into RL framework, and then RL-based scheduling, we first briefly explain BS, which is an important component of wireless communication. 5G and 6G massive MIMO concept aims to enhance throughput and optimize the energy efficiency of the wireless communication system [38].

BSs can be considered 5G cell internet towers. The number of cellular BSs keeps increasing due to the increasing demand for cellular devices. Moreover, a rise in cell phone users for data-heavy operations results in a strain on the existing towers. So, more BSs can help to enhance transmission between devices and cell tower antennas. The 5G technology employs mmWave signals that do not cover large areas as in 3G/4G networks. Therefore, specialized BSs are required to tackle the 5G mobile traffic.

### 4.1. Problem Statement

An MU-MIMO system in uplink configuration has a single BS that has $Rx_{M}$ receive antennas and *k* users each with a single transmit antenna as shown in Figure 2. When $\left(k>Rx_{M}\right)$, then BS needs to choose a group of users to allow them for transmission, and the selected group of users should be equal to available receive antennas $\left(Rx_{M}\right)$ at BS.

This scenario requires a user scheduling mechanism at BS before allowing the user to transmit. We consider the Rayleigh fading channel, and at each TTS, *N* users may be scheduled to uplink, while $N\leq Rx_{M}$. The data received by the BS can be written as given in Equation (Equation 2) [3].
(2)y=Hx+n
where $H\left(Rx_{M}\times N\right)=\sum_{i=1}^{K}\sqrt{P_{i}}h_{i}x_{i}$ is Rayleigh fading channel and $x={\left[x_{1},x_{2},...,x_{k}\right]}^{T}$ and ${\left[.\right]}^{t}$ denotes the transpose. While $x_{i}\in C^{N\times 1}$ is the transmitted signal, $P_{i}$ large-scale received power of one of the selected users, respectively, and $n∼N(0,\sigma^{2})$ is the AWGN.

### 4.2. Problem Formulation

As discussed in Section 3, an RL problem needs to be modeled as an MDP, that includes states, actions, and rewards.

**State**—Each underlying problem has a state or set of states that an RL agent may visit/explore. We map a state at time *t* as the combination of multiple parameters. First, we consider that each candidate user can determine its transmit beamforming vector using CSI, which is locally available [39]. Secondly, we use the combination of SINR, Gram Schmidt Orthogonalization, and SLNR as given in Equation (Equation 3). Each parameter is weighted equally, normalized between zero and one, and works as prior information for the BS. The former information is available to each user while the latter is received through BS.
(3)$$s_{t}=\left(SINR_{m},g_{m},SLNR_{m}\right)$$
where $SINR_{m},SLNR_{m}$ and $g_{m}$ indicate values of SINR, SLNR and Gram Schmidt Orthogonalization, respectively, for $m_{th}$ user form a state at time *t*, may be calculated as given in Equation (Equation 4) according to [19], Equation (Equation 5) according to [30] and Equation (Equation 8), respectively.
(4)$${\mathrm{SIN}R}_{m}=\frac{{∥h_{m}w_{m}∥}^{2}}{\sigma^{2}+\sum_{g(i)\in G}∥h_{m}w_{g(i)}∥}$$
where *G* is the subgroup of selected users, while $h_{m}$ and $w_{m}=h^{*}{\left(hh^{*}\right)}^{-1}$ are channel and precoding vector of $m_{th}$ user, respectively.
(5)$${\mathrm{SLNR}}_{m}=\frac{{∥h_{m}w_{m}∥}^{2}}{\sigma_{m}^{2}+\sum_{m\neq i}{∥h_{m}w_{m}∥}^{2}}$$
where
(6)$$w_{m}\propto {\mathrm{eigenvector}}_{\lambda_{max}}\left({\left(\sigma_{m}^{2}I+H_{m}^{*}H_{m}\right)}^{-1}h_{m}^{*}h_{m}\right)$$The $\lambda_{max}$ indicates the maximum SLNR of a user.To calculate $g_{m}$ for all users, the component of $h_{m}$ orthogonal to subspace spanned by $\left[g_{1}....g_{i-1}\right]$ as in [32].
(7)$$g_{m}=h_{m}-\sum_{j=1}^{i-1}\frac{h_{m}*g_{j}^{*}}{{∥g_{j}∥}^{2}}*g_{j}$$
(8)$$=h_{m}\left(1-\sum_{j=1}^{i-1}\frac{g_{j}*g_{j}^{*}}{{∥g_{j}∥}^{2}}\right)$$When i=1, then $g_{m}=h_{m}$.**Action**—Each user (agent) has to choose a resource block to transmit its data to BS. Therefore, an action is chosen by an agent *m* at time step *t* and can be written as given in Equation (Equation 9) as
(9)$$a_{t}=\left(U_{1,rb_{1}},U_{2,rb_{1}},...,U_{m,rb_{n}}....,U_{k,rb_{z}}\right)$$
where $U_{m,rb_{n}}$ represents that $m_{th}$ user selects $n_{th}$ resource block for data transmission.**Reward**—The natural objective of any user selection technique is to enhance the system capacity and optimal utilization of available radio resources. A reward function defines the goal in a RL problem. In every time step, the environment feedbacks to the RL agent a number as a reward. The agent’s goal is to maximize the total reward it receives over the long run. The reward or feedback thus defines the good and bad actions for the agent [6]. Sumrate is the metric used to indicate system performance in MU-MIMO systems. Therefore, the reward for the RL agent will be the aggregated sum-rate for all selected users as given in Equation (Equation 10) according to [28].
(10)$$R=C_{sum}=\sum_{j\in G}log_{2}\left(1+SINR_{j}\right)$$
where *G* is the group of users selected for transmission.

After having formulated an MU-MIMO user scheduling problem into a MDP by defining states, actions, and rewards, next we move toward the solution using the Bayesian RL method. The advantage of a Bayesian RL technique is that an agent can use the initial knowledge available in Equation (Equation 3), helping an agent learn and converge faster than classic RL approaches. Equation (Equation 11) needs to be solved for Bayesian RL.
(11)$$V^{\pi *}\left(x,b\right)=\max_{a}\sum_{x^{{}^{\prime }}}Pr\left(x^{{}^{\prime }}|x,b,a\right)\left[X_{r}^{{}^{\prime }}+\gamma V^{\pi *}\left(x^{{}^{\prime }},b_{xax^{{}^{\prime }}}\right)\right]$$
where the state as defined in Equation (Equation 3) is represented with *X* and distribution over the unknown θ is with *b*, respectively. The $Pr(x^{{}^{\prime }}|x,b,a)$ is used for transition probability. The probabilities may also be utilized as the posterior distribution P(θ|x) by employing θ given the initial information as given in Equation (Equation 1) and rewritten below in Equation (Equation 12).
(12)$$\begin{array}{c} P\left(\theta |x\right)=\frac{P(x|\theta )P(\theta )}{P(x)} \end{array}$$

We considered P(x|θ) and P(θ) as the Bernoulli and Beta distribution, respectively. As P(x|θ) and P(θ) are Bernoulli and Beta distributions, respectively, and since Beta distribution is a conjugate prior to the Bernoulli distribution, therefore, P(θ|x) is considered as Beta distributed. This indicates that when the aggregated sumrate increased after inclusion of a user, then (P(θ|x)) should be Beta(α + 1, β). Similarly, when the aggregated sumrate decreases after inclusion of a user, then the (P(θ|x)) should be Beta(α, β + 1).

To address the exploration–exploitation issue, we use Thompson Sampling [40]. It is a scheme for decision-making problems, and actions are performed sequentially to keep a balance between known actions (exploitation) and exploring new actions (exploration). In Thompson Sampling, the probability P(θ|x) is sampled from the prior. Then, the user with the highest sampled probability is moved to the group of users who will be allowed for transmission by the BS in each TTS.

### 4.3. RL-Based User Scheduling

This subsection focuses on the proposed RL-based user scheduling scheme. The detailed implementation of the methodology is shown in Figure 3 and a step-by-step explanation is given below.

Each user determines its transmit beamforming vector to quantify the amount of interference from the other users and resource block for data transmission as defined in Equation (Equation 9).As we are using the Rayleigh fading channel, which is to say that the channel gain from any of $Rx_{M}$ antennas to a user is described by ZMCSCG RV and this becomes a model that is suitable for narrow-band networks functioning in non-line-of-sight scenarios [41].BS After receiving feedback from users in the form of a priority resource block and amount of interference, calculate (SINR,SLNR,g) at each time TTS. Following BS selects the subset of users to equal the antenna number at BS to allow users for transmission based on all estimated information.After selecting a group of users, the aggregate sum-rate is calculated according to Equation (Equation 10). This sum-rate acts as feedback (reward), and based on the obtained reward for the selected users; each user can choose the most suitable resource block for transmission, which results in optimal utilization of available resources and enhancement in system capacity.

## 5. Results

This section reports the simulation results to show the performance of the proposed methodology. For simplicity, consider that each user has a single antenna and a BS with four and five receive antennas. Extension to more transmit and receive antennas may be studied in [42]. We consider a uniform transmit power for all data streams. The proposed scheme is compared with the Random, Low Complexity (LOWC) algorithm, SINR, and SLNR-based scheduling techniques for the Rayleigh fading channel. The performance metric we considered for evaluating an algorithm is the aggregated sum-rate of all the selected users when the BS has four receive antennas, then the number of users in the subset of the selected users will be four.

The first result shown in Figure 4 indicates the result when the user randomly selects a resource block for data transmission without considering the overall system capacity and other users’ priorities. We can see that the learning is mainly imbalanced and results in degraded performance. This is because the agent chooses action randomly and does not learn the environment’s dynamics. The experiment was conducted for 30 users, each with a single transmit antenna and four receiving antennas at BS.

On the contrary, in Figure 5, when the users consider the aggregated reward, which is the sum rate of all selected users, the overall performance is enhanced due to faster learning and convergence. Now, the agent is not taking actions randomly. Instead, it is continuously learning from the environment. At each time slot, a subset of users is selected for service, and the sum rate (reward) for the selected group of users is computed. Each time the better value of the sum rate over the previous one serves as positive feedback, and the agent learns that the chosen action was correct. While in case of negative feedback, the agent avoids choosing that particular action in that specific situation (state) again. The result is obtained for configuration four receiving antennas at BS and 20 users, each with a single antenna.

Furthermore, in Figure 6, we have presented a performance comparison in terms of sum-rate to demonstrate the feasibility of the RL method over the other methods. We have considered 30 users, each with a single antenna and five receiving antennas at BS. It is evident that the proposed technique performs better than the others, while the random algorithm performs worst.

Furthermore, we have found that the selection of the first user in the SLNR and SINR scheduling method (i.e., the user with maximum channel gain) is not realistic. We have investigated how the first user selection can be made arbitrary. Even a random choice of the first user will not affect the aggregated sum-rate performance of the SLNR and SINR algorithms.

In Figure 7 and Figure 8, we considered the performance of the proposed method and state of the algorithms referred as ALGO-1, ALGO-2 and ALGO-3 in Figure 7 and Figure 8 were presented in [33,34,35], respectively. The work in [33] uses the POMDP method for downlink transmit beamforming at BS equipped with multi-antennas. While a binary FPA is employed in [34] for both antenna and user scheduling to obtain sum-rate performance with comparatively less computational complexity. The framework in [35] applies a joint infrastructure for a hybrid precoder, and user scheduling is proposed.

In Figure 7, we considered ten candidate users, each with a single transmit antenna and four receive antennas at BS. At the same time, we increased the total number of candidate users to 20 in Figure 7 while keeping other parameters the same. We can quantify that the RL-based method obtains about 20% higher sum-rate performance than that of the ALGO-3 and quite significantly better performance over the other two schemes.

## 6. Conclusions

We have considered the user selection problem in an MU-MIMO system and introduced the multi-agent RL methodology as one of the solutions. We first modeled the user scheduling problem as MDP by defining states, actions, and rewards. We developed an optimal scheduling policy by optimally selecting a group of users for transmission with the help of RL.

The simulation results demonstrate that the proposed methodology provides a significant performance enhancement and indicates that AI-based techniques could play a vital role in communication systems. We have conducted experiments using four and five receive antennas at BS for 10, 20, and 30 users. A 20% higher sum-rate performance of the proposed scheme is reported for 10 and 20 users, while it achieves a slightly better sum-rate for 30 users and takes around 500 fewer episodes for convergence.

## Figures and Tables

**Figure 1 sensors-22-08278-f001:**
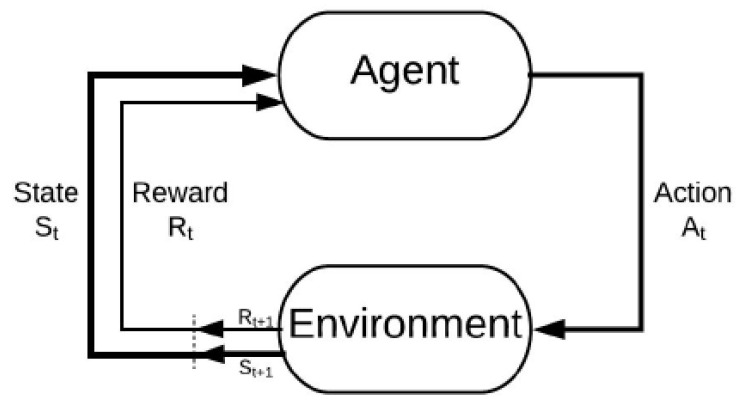
The reinforcement learning problem.

**Figure 2 sensors-22-08278-f002:**
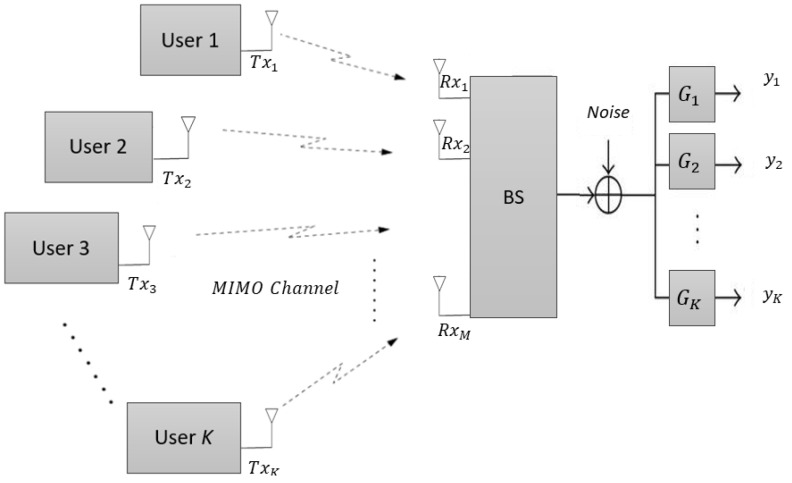
MU-MIMO uplink system with *K* users each with a single antenna and a BS with $Rx_{M}$ receive antennas.

**Figure 3 sensors-22-08278-f003:**
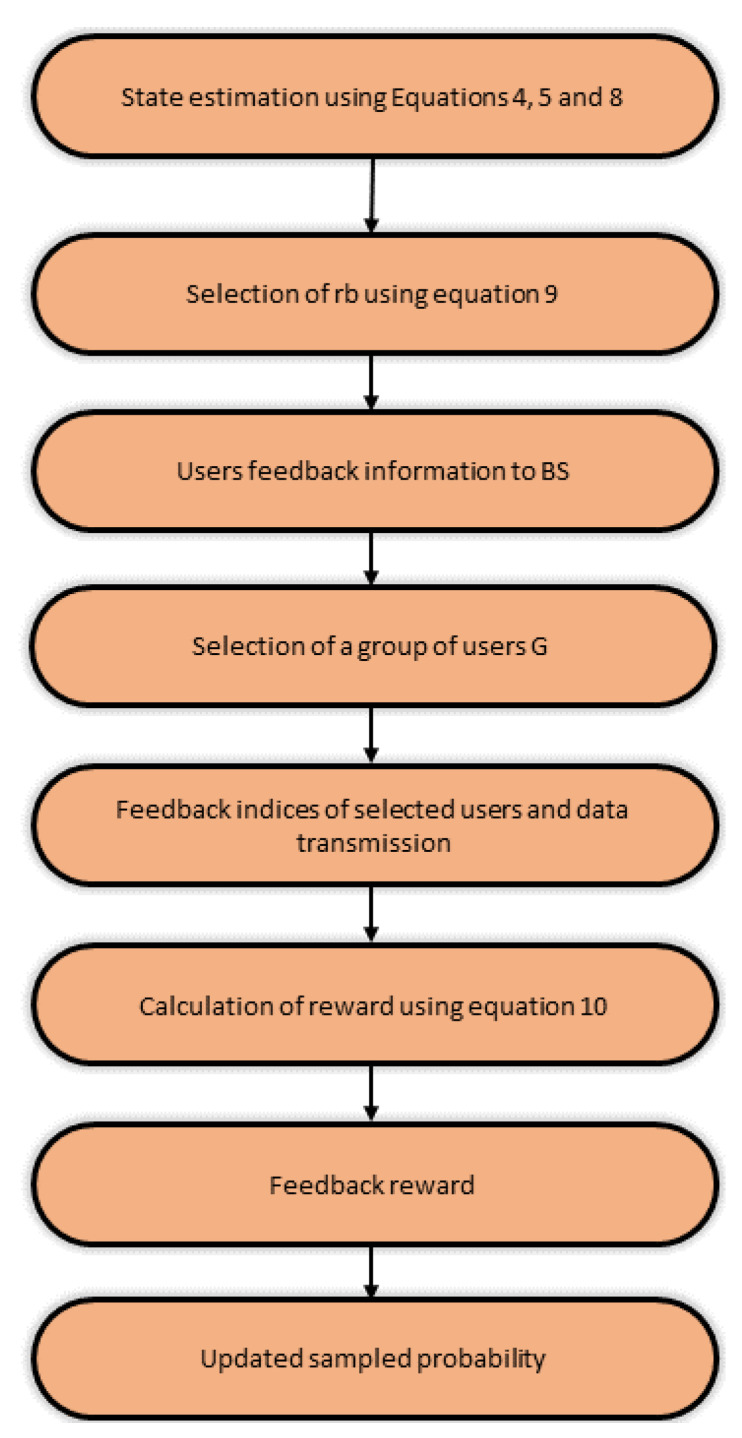
Flowchart of scheduling mechanism.

**Figure 4 sensors-22-08278-f004:**
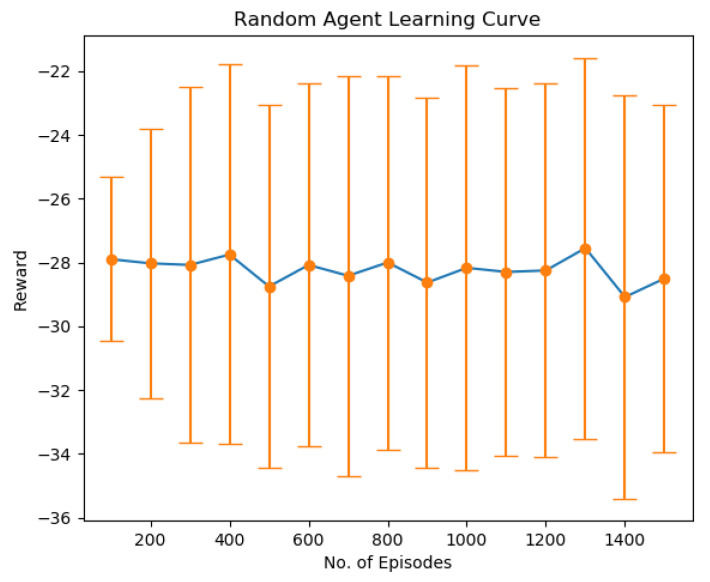
Sumrate performance of Random algorithm.

**Figure 5 sensors-22-08278-f005:**
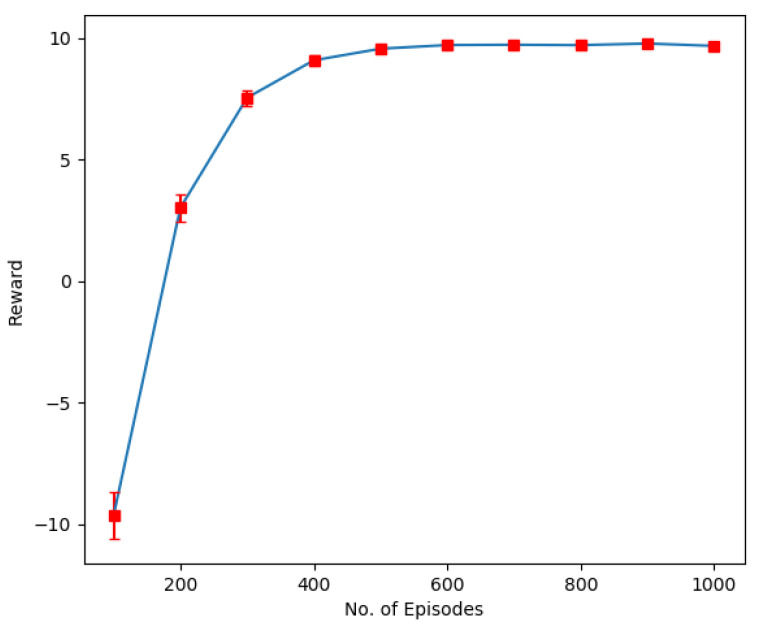
Sumrate performance of RL algorithm for 30 users, four receive antennas at BS and one transmit antenna at each user.

**Figure 6 sensors-22-08278-f006:**
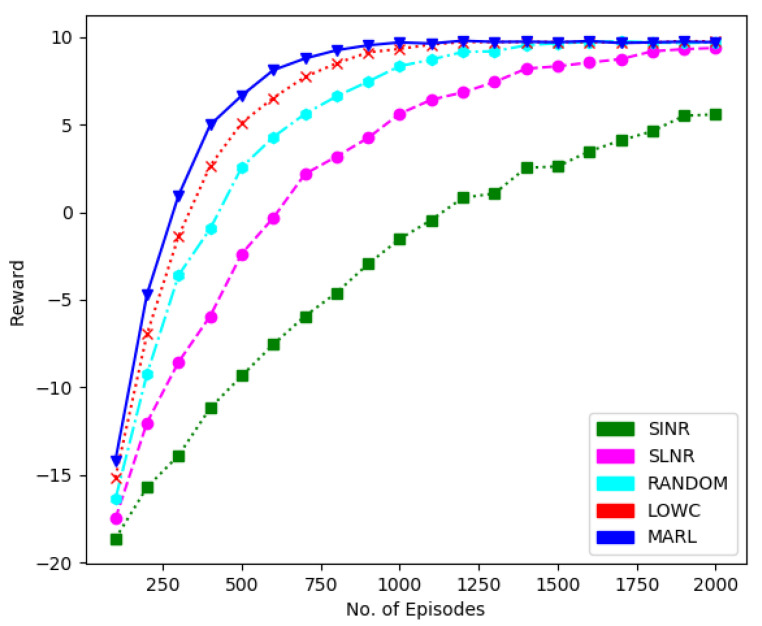
Sumrate performance comparison of different techniques for 30 users, five receive antennas at BS and one transmit antenna at each user.

**Figure 7 sensors-22-08278-f007:**
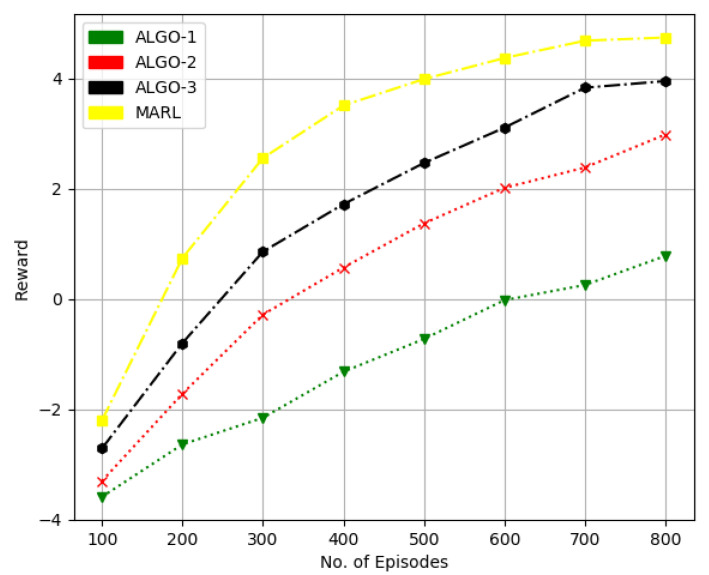
Performance comparison in terms of sumrate for 10 users, four receive antennas at BS and one transmit antenna at each user.

**Figure 8 sensors-22-08278-f008:**
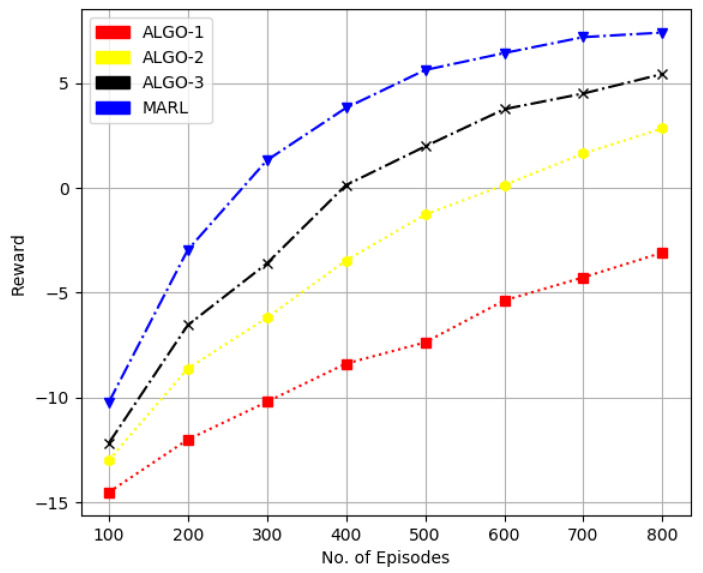
Performance comparison in terms of sumrate for 20 users, four receive antennas at BS and one transmit antenna at each user.

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
