# Peer review of "Optimal User Scheduling in Multi Antenna System Using Multi Agent Reinforcement Learning"

_sensors, 2022, doi:10.3390/s22218278_

Round 1
Reviewer 1 Report
This paper proposed a MU-MIMO scheduling with a reinforcement learning algorithm. This paper does not seem to be prepared well. The presentation needs to be significantly improved while English itself needs to be proofread by a native speaker. Among other things, the authors do not seem to present the essential existing works in MU-MIMO scheduling and fail to provide a proper comparison with the state of art methods.
Author Response
First Reviewer
Reviewer’s general comment
This paper proposed MU-MIMO scheduling with a reinforcement learning algorithm. This paper does not seem to be prepared well.
Authors’ answer
We are grateful for the chance that the reviewer provided us to improve the presentation and quality of our work. Please see below our response to each technical comment.
Reviewer’s question R1.1
The presentation needs to be significantly improved while English itself needs to be proofread by a native speaker.
Authors’ answer to R1.1
We are really thankful to the reviewers for carefully reviewing our work. We agree with the reviewer that the document needs improvement.
We have carefully revised the whole article to address all typos and grammar mistakes.
Reviewer’s question R1.2
Among other things, the authors do not seem to present the essential existing works in MU-MIMO scheduling and fail to provide a proper comparison with the state of art methods.
Authors’ answer to R1.2
The reviewer has highlighted two important points. One is related to literature and the other is related to the result section.
As for the first part of the comment concern, we have revised the entire related work section and have added existing work from 2021 and 2022 as advised by the reviewer. Please see the text in red color in the related work section
Regarding the second part of the comment, We have added more results in the updated version to improve the quality of the result section. Please see Figures 7 and 8 in the revised version.
Reviewer 2 Report
The authors in this work propose an idea about RL in Wireless that it is an important problem.
So, the authors should add more clarification of the model:
1. Add an algorithm explicitly for the RL
2. HOW TO COMPUTE THE REWARD DYNAMICALLY
3. ADD MORE EXPLANATION ABOUT MORE DETAILS FOR THE BS IN THE NEXT NETWORK 5G OR 6G?
Author Response
Second Reviewer
Reviewer’s general comment
The authors in this work propose an idea about RL in Wireless that it is an important problem.
So, the authors should add more clarification of the model:
Authors’ answer
We are very grateful to the reviewer for encouraging comments. Please see below the response to each technical comment.
Reviewer’s question R2.1
Add an algorithm explicitly for the RL
Authors’ answer to R2.1
Thank you for your valuable suggestions. We have added RL algorithms in the revised version.
Reviewer’s question R2.2
HOW TO COMPUTE THE REWARD DYNAMICALLY
Authors’ answer to R2.2
We are grateful to the reviewer for his valuable comment.
We have tried to better explain the reward function in the revised version. We also briefly explain it here.
A reward function defines the goal in an RL problem. In every time step, the environment sends to the RL agent a single number, a reward. The agent’s objective is to maximize the total reward it receives over the long run. The reward or feedback thus defines what are the good and bad events for the agent. In our example (MU-MIMO scheduling problem), we consider aggregated sum-rate of all selected users as the reward and it is computed using Equation 10.
Reviewer’s question R2.3
ADD MORE EXPLANATION ABOUT MORE DETAILS FOR THE BS IN THE NEXT NETWORK 5G OR 6G?
Authors’ answer to R2.3
This is an important point and we are grateful to the reviewer for highlighting it. We have added the necessary material regarding BS for 5G networks in the revised version. Please refer to the highlighted text in red color in the system model section.
Reviewer 3 Report
Thank you for your effort and interesting work.
I have some comments that must be considered in the modified manuscript.
----------------------------------------------------------------------------------
1) I see both (Abstract) and (Conclusion) are descriptive. It is better for (Abstract) to have at least one numerical value of main results. The (Conclusion) must include many numerical values of your findings.
2) Explain both abbreviations: RL (Reinforcement Learning) and MDP in your (Abstract).
3) You used RL methodology. Why? State its advantages over at least another methodology.
4) Please add a block diagram (or any means) to describe the procedure of the mathematical model. This helps in understanding the sequence of equations.
5) Most of equations need references.
6) Equation 10 (looks like Shanon equation) gives the Reward. What is the physical meaning of the term (SINR). I see its mathematical representation in Eq. 5.
7) Figures 3+4 are poor in resolution. Please, use larger fonts in numbering both axes and axes titles.
8) I do not see any comparison with any previous work. OR: at least, tell us how do you judge correctness of your findings.
9) What is the difference between results in Fig. 3 and in Fig. 4? What is the physical meaning of negative reward.
10) In your (Abstract), you write
Simulation results show that the RL methodology enhances the performance significantly.
What do you mean by performance? is it the MIMO system performance? and what is the performance evaluation parameter that you considered to say (enhancement)?
11) We are in 2022. I did not find any reference in 2022 and found only one reference in 2021. Please update.
Author Response
Third Reviewer
Reviewer’s general comment
Thank you for your effort and interesting work.
I have some comments that must be considered in the modified manuscript
Reviewer’s question R3.1
I see both (Abstract) and (Conclusion) are descriptive. It is better for (Abstract) to have at least one numerical value of the main results. The (Conclusion) must include many numerical values of your findings.
Authors’ answer to R3.1
Thank you for your significant suggestion. We have incorporated your advice in the revised version.
Reviewer’s question R3.2
Explain both abbreviations: RL (Reinforcement Learning) and MDP in your (Abstract).
Authors’ answer to R3.2
Thank you for highlighting this issue. We have carefully revised the whole document and have corrected all abbreviations.
Reviewer’s question R3.3
You used RL methodology. Why? State its advantages over at least another methodology.
Authors’ answer to R3.3
The reviewer has raised an important question.
We have added a better explanation and more precisely the advantage of the RL method over the state-of-the-art.
Reviewer’s question R3.4
Please add a block diagram (or any means) to describe the procedure of the mathematical model. This helps in understanding the sequence of equations.
Authors’ answer to R3.4
We are thankful to the reviewer for pointing our this limitation. We have added a flow chart that presents a step-by-step working of the proposed methodology.
Reviewer’s question R3.5
Most of the equations need references.
Authors’ answer to R3.5
We agree that references for some equations were missing in the original manuscript. We have added the references to some equations where required on the reviewer's advice.
Reviewer’s question R3.6
Equation 10 (looks like Shanon equation) gives the Reward. What is the physical meaning of the term (SINR)? I see its mathematical representation in Eq. 5.
Authors’ answer to R3.6
The reviewer is right about his/her observation. Equation 10 is the standard equation that is used to calculate the system capacity and we use the term aggregated sumrate of selected users to present the performance of a scheduling algorithm. SINR stands for Signal to Interference plus Noise ratio. As we are dealing with multiple users there the inter-user interference is also taken into account.
Reviewer’s question R3.7
Figures 3+4 are poor in resolution. Please, use larger fonts in numbering both axes and axes titles.
Authors’ answer to R3.7
Thank you for pointing out this issue.
We have corrected the mistake and now axes numbers and titles are readable.
Reviewer’s question R3.8
I do not see any comparison with any previous work. OR: at least, tell us how you judge the correctness of your findings.
Authors’ answer to R3.8
Yes, the reviewer is right that a necessary comparison was missing in the original draft. We have added more results in the updated version to improve the quality of the result section. Please see Figures 7 and 8 in the revised version.
Reviewer’s question R3.9
What is the difference between the results in Fig. 3 and in Fig. 4? What is the physical meaning of negative reward?
Authors’ answer to R3.9
Thank you for your comment. The comment is quite relevant and the presentation for both figures was not good in the original version. We have improved the presentation and also explain here.
Figures 3 and 4 show the learning outcome against two different algorithms. Figure. 3 shows the learning of a random agent and we can see that agent is not learning very well as it chooses the action randomly. On the other hand, the RL agent continuously learns the dynamics of the environment and it is evident from the learning curve shown in figure.4 of the original version.
Regarding the second part of the Reviewer's comment: in general, the reward may be positive or negative. It is the feedback that tells the agent whether its chosen action is good or not. In case of a negative reward, an agent learns that the chosen action was not good and next time an agent should avoid selecting that action in a particular state, and the whole process refers to the learning of the agent. The ultimate goal of an agent is to maximize the aggregated reward.
Reviewer’s question R3.10
In your (Abstract), you write Simulation results show that the RL methodology enhances performance significantly. What do you mean by performance? is it the MIMO system performance? and what is the performance evaluation parameter that you considered to say (enhancement)?
Authors’ answer to R3.10
This is an important and necessary concern that the reviewer has asked. We have used aggregated sum-rate of selected users as a performance metric to evaluate the performance of a scheduling algorithm. The aggregated sum-rate of served users in a time slot indicates the MU-MIMO system performance (capacity). We have also improved our presentation about performance metrics and evaluation of performance in the revised version.
Reviewer’s question R3.11
We are in 2022. I did not find any reference in 2022 and found only one reference in 2021. Please update
Authors’ answer to R3.11
Thank you for pointing out this important issue. We agree with the reviewer that most of the references in the original draft were not from recent years.
We have revised the entire related work section and have added existing work from 2021 and 2022 as advised by the reviewer. Please see the text in red color in the related work section
Reviewer 4 Report
The article Optimal User Scheduling in Multi Antenna System using Multi Agent Reinforcement Learning is focused on transmission scheduling in multiple MIMO systems. The authors proposed user’s selection method based on multi-agent reinforcement learning technique. The method uses an optimal scheduling policy by optimally selecting a group of users to be scheduled for transmission given the channel condition and resource blocks at the beginning of each time slot.
First, the language throughout the entire paper should be improve, as I was able to notice various mistakes, typos and errors in the text. For example: line 146 – “An MU-MIMO…”, “… that hash Rxm…”, line 162 – “.. Um,rbn represent…”, line 232 - …” and indicate that AI…”, etc. Word “Equation” is used with capital letter as well as small letter “equation” in the text.
Next, there are several abbreviations in the text, which are not described and included in the list at the end of the paper, for example: TTS, SVD, DPC, etc.
Some of the parameters and symbols in Eqs. (4), (5), (7), (8) are not defined and properly described.
Line 170 – “…obeys the Bernoulli and Beta distribution respectively.” – there is no proof that both distributions should agree Bernoulli and Beta distributions.
The results provided in Section 5 are brief and poorly presented. More intensive simulations for various different scenarios should be provided. Moreover, better presentation of results and their description is necessary. Better comparison with existing methods should be provided.
The list of references contains a lot of outdated references and it should be updated.
Author Response
Fourth Reviewer
Reviewer’s general comment
The article Optimal User Scheduling in Multi Antenna System using Multi Agent Reinforcement Learning is focused on transmission scheduling in multiple MIMO systems. The authors proposed user’s selection method based on multi-agent reinforcement learning technique. The method uses an optimal scheduling policy by optimally selecting a group of users to be scheduled for transmission given the channel condition and resource blocks at the beginning of each time slot.
Author's response:
We are really grateful to the reviewer for his/her valuable feedback and suggestions. These comments helped us to further improve the quality of our article. Please see below the detailed response to each comment. Once again thank you for your time and suggestions.
Reviewer’s question R4.1
First, the language throughout the entire paper should be improve, as I was able to notice various mistakes, typos and errors in the text. For example: line 146 – “An MU-MIMO…”, “… that hash Rxm…”, line 162 – “.. Um,rbn represent…”, line 232 - …” and indicate that AI…”, etc. Word “Equation” is used with capital letter as well as small letter “equation” in the text.
Authors’ answer to R4.1
Thank you for carefully reading our manuscript. We have not corrected the mentioned mistakes but also revised the whole manuscript. We have also used the word ‘Equation’ in the revised manuscript for all equations.
Reviewer’s question R4.2
Next, there are several abbreviations in the text, which are not described and included in the list at the end of the paper, for example TTS, SVD, DPC, etc
Authors’ answer to R4.2
Thank you for pointing out these mistakes. Indeed, these were important terms and their description were missing. We have corrected all abbreviations in the revised version.
Reviewer’s question R4.3
Some of the parameters and symbols in Eqs. (4), (5), (7), (8) are not defined and properly described.
Authors’ answer to R4.3
We are sorry for this inconvenience. Although few terms were defined in previous equations, the reviewer is right that many terms used in equations 4 to 8 were missing. We have carefully revised all equations and tried to define all terms used in any equation.
Reviewer’s question R4.4
Line 170 – “…obeys the Bernoulli and Beta distribution respectively.” – there is no proof that both distributions should agree Bernoulli and Beta distributions
Authors’ answer to R4.4
This is an important consideration. The reviewer was right about his concern as our statement was confusing. We have modified the text in the revised version and also explained it here.
We did not assume, in fact, we used Beta distribution for the prior P(θ) since it is a conjugate prior to the Bernoulli distribution. This means that if the likelihood function P(x∣θ) is Bernoulli distributed and the prior distribution P(θ) is Beta distributed then the posterior P(θ∣x) will also be Beta distributed.
Reviewer’s question R4.5
The results provided in Section 5 are brief and poorly presented. More intensive simulations for various different scenarios should be provided. Moreover, a better presentation of the results and their description is necessary. A better comparison with existing methods should be provided.
Authors’ answer to R4.5
We agree that all points highlighted by the reviewer related to the Result section are correct. We have added more results in the updated version to improve the quality of the result section. Please see Figures 7 and 8 in the revised version.
Reviewer’s question R4.6
The list of references contains a lot of outdated references and it should be updated.
Authors’ answer to R4.6
Thank you for pointing out this important issue. We agree with the reviewer that most of the references in the original draft were not from recent years.
We have revised the entire related work section and have added existing work from 2021 and 2022 as advised by the reviewer. Please see the text in red color in the related work section
Round 2
Reviewer 2 Report
The authors are improved the revised version by reply to our comments. If the paper is accepted, the authors add the next references for the readers:
Robust Enhancement of Intrusion Detection Systems using Deep Reinforcement Learning and Stochastic Game, IEEE Transactions on Vehicular Technology, 2022, DOI: 10.1109/TVT.2022.3186834.
Game Model For Dynamic Cell Association Of Macro-User In Two-Tier Cellular Networks. In The Proceedings Of The International Conference IEEE Global Communications (Globecom2018) 9-13 December 2018, Abu Dhabi, UAE.
Reviewer 4 Report
The authors have provided sufficient revisions in the paper and they revised their paper according to the reviews. I recommend to accept the paper.